# The Impact of Time on Nutrition and Health Claims on the Irish Marketplace

**DOI:** 10.3390/foods11182789

**Published:** 2022-09-09

**Authors:** Stefanie Marisa Offe, Loanne Bebin, Fiona Lalor

**Affiliations:** 1School of Agriculture and Food Science, University College Dublin, Belfield, D04 V1W8 Dublin, Ireland; 2School of Biomedical Sciences, University of Ulster, Coleraine BT52 1SA, UK

**Keywords:** nutrition claims, health claims, Irish retail market, EU Regulation 1924/2006

## Abstract

Since the implementation of Regulation (EC) No 1924/2006 on nutrition and health claims (NHCR) made on food, only 261 health claims have been authorised, suggesting that the regulation creates challenges for the food industry. This study looks at the prevalence of nutrition and health claims labelled on food on the Irish market. Specifically, we compared their prevalence in 2009 with those present on the marketplace in 2022. Food labels of defined food categories were examined in three nationwide supermarkets in Ireland, and data of claims made on these labels were collected. A series of statistical analyses was conducted to compare the results from 2021 with those of 2009. In 2021, around half of the examined products (52.8%) displayed at least one nutrition claim, showing no significant difference with the data collected in 2009 (53.5%). Individual categories, however, did exhibit changes between the two time points. The prevalence of health claims on food has decreased from 21.6% in 2009 to 10.5% in 2021. There will always be a time lag between when a regulation is enacted and what its true impact can be measured. This study provides these data for the impact of time on EU Regulation 1924/2006 on the Irish market.

## 1. Introduction

In the aftermath of several food scares in the European Union (EU), such as the Bovine Spongiform Encephalopathy (BSE) crisis in the early 1990s and the Belgium Dioxin crisis in 1999, EU regulators have put an increased emphasis on food safety whilst striving to guarantee the highest consumer protection [1]. Moreover, due to our changing lifestyles, including our changing dietary patterns, non-communicable diseases (NCDs) such as cardiovascular diseases, diabetes mellitus type 2, and other nutrition-related diseases, cause more death each year than any other disease [2]. Data published by the World Health Organisation in 2021 indicate that 71% of all deaths globally are attributable to NCDs, and that each year, more than 15 million people between the ages of 30 and 69 die from NCDs [3]. Given the established relationship between overweight/obesity and NCDs, consumers have become increasingly interested in “optimal nutrition” rather than “adequate nutrition” [4]. In other words, the focus of nutrition has shifted towards the potential to promote health and prevent diseases, either through improving physical or mental conditions or by reducing risk factors attributed to diseases [5].

Due to increasing nutrition awareness, and partially due to the increased financial burden of rising healthcare costs from NCDs, for example, or the aging society [6], the EU commenced a series of measures aimed to lower the rise of nutrition-related diseases [7]. In the late 1990s and early 2000s, nutrition and health claims were being used in the food and drink industry all over the world. Food business operators (FBOs) endorsed nutrition and health labelling in order to provide nutritional information to consumers [8] and further use those messages on food packages as communication tools and as brand marketing strategies [9]. This was not provided in any meaningful regulatory way and generated an unlevel marketplace because the consumer purchasing choice can typically be strongly influenced by the perception of food labels [9], and these were not controlled in an equitable way. In order to manage this industry practice and protect consumers by harmonising all voluntary statements of nutrition or health-related effects on pre-packaged food across the European Union Member States, the EU implemented the first piece of legislation on the subject [10,11]. European Regulation 1924/2006 on nutrition and health claims made on food (NHCR) came into force in 2007. Although all European countries are subject to the rules of the regulation, enforcement remains the responsibility of each Member State. This inevitable disparity in uniformity of enforcement can be a source of difficulties for the food and drink industry [12]. Prior to the implementation of EC Regulation 1924/2006, only a limited legal framework controlling claims made on food existed, and even then, only in certain EU Member States. As a result, prior to the publication of the legislation, non-scientifically proven or exaggerated claims were made, resulting in mistrust and consumer confusion [13].

The objectives of this regulation are to guarantee the highest standards of consumer protection, to encourage food-related innovation, and to ensure the free movement of goods throughout the EU [14]. To meet this requirement, each claim must comply with scientific substantiation and must not be incorrect, misleading, or confusing [15]. Although this regulation came into effect in 2007, it is imperative that regulators and the food industry understand its impact. A study was conducted by Lalor et al. [16] in 2009, which was a snapshot of nutrition and health claims found on the Irish marketplace at the time. This paper compares the marketplace then with data collected in 2021 to assess the impact of time on the use of nutrition and health claims by the food and beverage industry in Ireland.

Article 2 of the NHCR defines a claim as “any non-mandatory statement or presentation which also includes graphics, symbols, or pictorials, that implies or suggests that any food possesses certain beneficial properties” [14]. The legislation creates two different categories of claims: nutrition claims (NC) and health claims (HC).

Article 2.4 of the NHCR defines a nutrition claim as “any claim which states, suggests, or implies that a food has identified beneficial nutritional properties due to:(a)The energy it provides, provides at a reduced or increased rate, or does not provide; and/or(b)The nutrients, or other substances it contains, contains in reduced or increased proportions, or does not contain” [13].

Until May 2022, the European Commission had authorised 30 nutritional claims, such as “low energy” and “source of omega-3 fatty acids”. This list of authorised nutrition claims included four comparative claims, such as “Increased…” or “light/lite” [13]. All nutrition claims and their conditions of use are given in the Annex of Regulation 1924/2006.

Article 2.5 of the NHCR additionally defines a health claim as “any claim that states, suggests or implies that a relationship exists between a food category, a food, or one of its constituents and health” [13].

The regulation further divides health claims into three types: Article 13.1 claims, Article 13.5 claims and Article 14 claims:

Articles 13.1. and 13.5 claims are also known as “functional claims” referring to:(a)“The role of a nutrient, or other substance in growth, development, and the functions of the body, or(b)Psychological and behavioural functions, or(c)Without prejudice to Directive 96/8/EC, slimming or weight-control, or a reduction in the sense of hunger, or an increase in the sense of satiety, or to the reduction of the available energy from the diet” [13].

Article 14 (a) claims are known as “Reduction of disease risk claims”, which associate a relationship between a foodstuff and the reduction in a risk factor of diseases [13,17,18].

Article 14 (b) claims have a special focus on children’s development [19]. Table 1 shows the differentiation between Nutrition and Health Claims and includes examples.

To utilise a nutrition or health claim on prepacked food, several procedures need to be followed. In accordance with EC Regulation 1924/2006, a nutrition claim can only be used when permitted and listed in the annex of Regulation 1924/2006. Additionally, the products on which the nutrition claim is being made must fulfil the requirements laid down by the regulation itself [14]. The list of permitted claims was updated in 2012 with the publication of Regulation (EC) 1047/2012. A change to this list can only be published through a comitology procedure at the initiative of the Commission [21].

In the EU, health claims are treated very differently from a regulatory perspective, and pre-market approval is required for their use. A food business operator (FBO) must submit a complete dossier of the food or ingredient, including the specification for its use, scientific evidence, and a proposal for the actual on-pack wording [22]. It is the task of the European Food Safety Authority (EFSA) to assess the evidence submitted by an applicant. The European Commission will then consider the opinion issued by the EFSA and publish the outcome of the application—either positive or negative [23,24]. An exemption exists for Article 13.1 health claims. In this case, upon publication of the Regulation in 2007, FBOs were invited to submit proposals to the competent authority of the Member States who forwarded it, after giving consent, to the European Food Safety Authority (EFSA). 

On 16th May 2012, the list of permitted health claims (Article 13.1), other than those referring to the reduction in disease risk and to children’s development and health, and their conditions of use was published in Commission Regulation (EU) 132/2012 [25]. Of the 4637 health claims applications received, supported by generally accepted scientific evidence, only 229 were authorized [26]. Currently, 1500 claims associated with the beneficial effects of botanicals have not yet been assessed [26]. Table 2 presents the number of assessed health claims and the number of authorized claims in each category [26].

Due to uncertainties in the application process or the perceived insufficient substantiation with scientific evidence required for claim substantiation, many applications have not received authorization [11]. In an attempt to assist the industry, the EFSA has published a variety of guidance documents for stakeholders which are revised routinely to ensure better clarification on the requirements of the approval process [28,29,30]. Among other aspects of the process, these guidance documents attempt to clarify issues such as requirements for the characterisation of food, the evidence required for scientific substantiation, the identification of pertinent human studies and the extrapolation of the results from a study group to the target population. Despite these guidance documents, however, after almost 15 years since the initial publication of NHCR, many challenges still exist.

Since the publication of the NHCR in 2006, the legislation resulted in several significant changes for companies wishing to use a claim. The administrative requirements became significantly higher. For example, ‘new’ health claims (those that were based on newly developed scientific evidence) were subject to an authorisation procedure and required relevant supportive evidence. One of the main reasons for the 2077 non-authorised health claim applications [26] was the lack of appropriate human studies [21]. Thus, the number of permitted claims has decreased compared with the prevalence of claims used before the implementation of the regulation [21].

Many agri-food operators claim that the regulation is inadequate, due to a lack of information and requirements that are considered far too difficult to achieve [21]. Partially due to this relatively low number of approved claims, the NHCR has been widely discussed and criticized. It has been the topic of various debates and studies [31] mostly focused on consumer perception and understanding as well as on purchasing patterns, property data, and innovation [10,23,32,33,34,35]. Other studies have concentrated on the prevalence of nutrition and health claims on the European market at a given time [8,36].

Limited research has been conducted on the prevalence of nutrition and health claims on a Member State’s market, examining the trend over time, particularly since the implementation of EC Regulation 432/2012.

For this reason, the present study aims to fill the gap in the existing literature examining the trend over time by evaluating the prevalence of nutrition and health claims on the Irish market in 2021 in comparison with 2009, as presented by Lalor et al. [16]. The research question of this study is: How much has the existence of nutrition and health claims on prepacked food of defined food categories on the Irish market changed over time?

## 2. Materials and Methods

The survey was conducted in Dublin City Centre, Ireland, to investigate the prevalence of nutrition and health claims on labels of packaged food products available on the Irish retail market. Food labels of different food categories were examined by hand. The data were collected in three of Ireland’s best-known supermarkets: Tesco, SuperValu, and Lidl. All chosen retailers are available nationwide and supply major brands. It was considered that they represented a national sample due to the market shares in Ireland.

The data collection was carried out in two parts. In July 2021, the data were collected in the first four food categories including:-Breakfast Cereals;-Yoghurt and Yoghurt Drinks;-Fruit Juices and Smoothies;-Frozen Fruit and Vegetables.

The second collection took place in October 2021 and concerned the following 4 product categories:-Bread and Bakery;-Soft Drinks;-Cheese;-Pasta and Rice.

These eight categories were chosen to match those in the study published 12 years earlier by Lalor et al. (2009) [16], to enable an accurate comparison.

The information that was collected is freely available in the public domain; therefore, ethical approval was not required. However, before collecting the data, approval was obtained from the supermarket managers of each store.

All food products in each category were listed either directly in a Google Form designed for this study and stored in a Microsoft Excel Sheet, or the labels of these products were photographed to enable completion of the Google Form afterwards. For the purpose of this study, the following information was taken from the product packaging:-1. Name/brand of product;-2. Manufacturer;-3. Food category;-4. Absence/presence of nutrition or health claims;-5. Type of claims;-6. Number of nutrition claims;-7. Exact text of nutrition claims;-8. Number of health claims;-9. Exact text of health claims.

Claims referring to the presence of additives such as “No additives” or “no artificial colour added”, or allergens such as “gluten free”, or “allergen free” were not taken into account, because they are not designated as health or nutrition claims in accordance with EU Regulation 1924/2006. Similar claims written in several places on the packaging were taken into consideration only once.

The data collected were entered in a Microsoft Excel spreadsheet, duplicate or triplicate entries were identified and excluded, and claims were checked against the NHCR to ensure that the spreadsheet did not include unauthorised claims.

The remaining data were transferred to the IBM^®^ SPSS Statistic program version 26 to be analysed. The overall frequency of claims (health and nutrition) and the frequencies in each food category were calculated. Furthermore, the number of claims, as well as the type of nutrients (fibre, sugar, fat, vitamin, etc.) highlighted on the packages were analysed.

Finally, chi-squared tests were carried out to compare the proportions of NC and HC on the marketplace in 2009 with those on the marketplace in 2021.

## 3. Results

### 3.1. Nutrition and Health Claims on the Marketplace in 2021

In this study the data of 1636 food products (*n* = 1636) were examined and analysed. The products were divided into the 8 food categories shown in Table 3. These eight categories were chosen to match those in the study published 12 years earlier by Lalor et al. (2009) [16] to enable accurate comparisons.

Overall, 54.1% (*n* = 885) of products examined carried at least one nutrition or health claim. As outlined in Table 4, there is a distinction between nutrition claims and health claims. From all products examined, 52.8% (*n* = 863) carried only nutrition claims, whereas 10.5% (*n* = 172) of the food products displayed only health claims on the packaging. Additionally, 9.2% (*n* = 150) carried at least one nutrition and one health claim.

#### 3.1.1. Results by Category

In accordance with Table 5, Breakfast Cereals was the category with the highest proportion of nutrition claims (81.2%) in 2021, followed by Yoghurt and Yoghurt Drinks (62.2%), Fruits Juices and Smoothies (60%), and Soft Drinks (58.2%). The categories of Cheeses as well as Pasta and Rice were those which displayed nutrition claims less frequently: 28.7% and 27.3%, respectively.

There was a disparity in the prevalence of health claims, ranging from 31.8% for Fruit Juices and Smoothies to 0% for Soft Drinks and Pasta and Rice. Detailed results are presented in Table 5.

#### 3.1.2. Type of Nutrient Involved

In this study, the proportions of nutrition claims across all food categories in terms of nutrient types were examined. These were divided into fat, protein, fibre, sugar, salt, and vitamins or minerals. Table 6 outlines the prevalence of the above-mentioned nutrients. 

The majority of the observed nutrition claims referred to vitamins or minerals, with 362 nutrition claims (24%) in which “Source of Vitamin C” accounted for the largest share. This was followed by 21% of food products (314 of the sample) making a fat content claim, and 20% stating the sugar content. Moreover, 19% referred to the fibre content. Figure 1 presents the complete results.

### 3.2. Comparison with Results Presented by Lalor et al., 2009

To compare the impact of time on the data collected, the percentages of the prevalence of nutrition and health claims were obtained from a study conducted in 2009 by Lalor et al. In order to conduct a chi-squared test and to analyse the trend over time, the overall percentage of nutrition and health claims, as well as category-specific, percentages were used.

As already mentioned, the category displaying the most nutrition claims in 2021 was Breakfast Cereals, whereas in 2009. most nutrition claims were found in the food category Frozen Fruits and Vegetables. Conversely, the category with the most health claims in the present study was Fruit Juices and Smoothies. Lalor et al. (2009) [16], however, identified most health claims within the food category Yoghurt and Yoghurt Drinks.

When comparing the results presented in Table 7, it is noticeable that the proportion of nutrition claims has evolved very differently across the categories compared with the results gathered in 2009. In the categories Fruit Juices and Smoothies, Frozen Fruits and Vegetables, and Cheeses, the number of nutrition claims has decreased, whereas it has increased for Yoghurt and Yoghurt Drinks, and Soft Drinks. The prevalence almost stagnated (less than 5% change) for Breakfast Cereals, Breads and Bakery, and Pasta and Rice products.

Looking at health claims, it is apparent that their use dropped in almost all categories except in the categories Frozen Fruits and Vegetables, in which it grew from 0% to 11.1%, and Fruit Juices and Smoothies where an increase of almost 3% is found (from 29% to 31.8%).

In the overall comparison of the total frequency of nutrition and health claims on food products on the Irish market, a stability for nutrition claims can be seen over time, whereas a decrease is visible for the prevalence of health claims.

The results of the chi-squared test, (testing whether a significant different exists between the prevalence of nutrition claims on food products on the Irish market between 2009 and 2021) was *p* = 0.736; hence, the *p*-value was above 0.05. Therefore, there is no significant difference in the use of nutrition claims labelled on food for all categories over time.

However, the prevalence of health claims on food products shows significant differences between the data obtained in 2009 and in 2021 (*p* < 0.001). These results are clearly demonstrated in Table 7.

## 4. Discussion

Nutrition and health claims displayed on prepacked food are important to help consumers make healthier food choices, whilst supporting the increased intake of specific nutrients. From a public health point of view, nutrition- and health-related information are pivotal to improving the health status of the public [37]. Regarding the awareness of the potential beneficial effects of food labelling, this study was conducted to determine how the existence of nutrition and health claims on pre-packaged food of eight defined food categories on the Irish market has changed over time.

The impact of time is apparent when the two types of claims are considered separately. In fact, the prevalence of nutrition claims on the Irish market has not changed over time, with 53.5% of the products carrying at least one nutrition claim in 2009 compared with 52.8% today, indicating that the use of nutrition claims has remained stable. However, the evolution of health claims is different. In 2009, 21.6% of the products carried a claim versus 10.5% today, which demonstrates a significant twofold decrease.

There are two potential explanations for why this has happened. The first is the alterations to our lifestyles and consumption habits; the second is the introduction of EU Regulation 1924/2006 in early 2007.

In the study conducted by Lalor et al. in 2009, the data were collected in September/October 2007, one year after the regulation came into force. At that time, food products with nutrition and health claims labelled prior to the application of the NHCR were allowed to be marketed until their expiry date, but no later than the end of July 2009 [14]. This means that, during this transition period, more nutrition and health claims were used on products without authorization or claims that were subsequently rejected. This could explain the decrease in the use of claims for the Fruit Juices and Smoothies, Frozen Fruits and Vegetables, and Cheeses categories. Indeed, the use of nutrition claims in the Cheeses category has decreased from 41% to almost 29%, although the dairy and dairy product consumption in Ireland has increased over the last two decades from 2000 to 2021 [38]. A similar decrease was seen in a study conducted in Serbia in 2021 [39]. When taking the different types of nutrition claims into account, no claims displaying “source of protein” were used in the study conducted by Lalor et al. in 2009, but in the examination in 2021, almost one-quarter (24%) of all nutrition claims within the Cheeses category referred to protein-related claims. The introduction of “protein” as the nutrient of choice on food packages and the desire of manufacturers to promote it on their products is reasonable, when considering that the majority of the Irish population think that eating dairy products such as cheese is a good way of adding protein into their daily diets [40].

Similarly, the consumption of fermented dairy products (yoghurt), has been very high across the European Union and also in Ireland, resulting in a successful market for nutrition claims on yoghurts [41]. Additionally, numerous studies have highlighted the beneficial health outcomes of yoghurt consumption, such as in reducing type 2 diabetes, weight gain, and obesity due to the reduction in body fat and reducing insulin resistance [42,43,44]. Furthermore, the beneficial changes in the gut microbiome due to probiotics and prebiotics have been outlined multiple times [45,46]. Together, this has led to increased yoghurt consumption, and claims on yoghurt are now considered a great communication tool [47]. Indeed, this study determined an increase of almost 7% in the prevalence of nutrition claims on yoghurts (55–62%).

Another surge in the use of nutrition claims from 45% to 58% was observed for soft drinks. The data show that the majority of nutrition claims in this category referred to the sugar content. This can be explained by the current health trend in reducing or limiting the consumption of sugar, in particular in the beverage sector, and due to the taxes on soft drinks that were implemented by several European countries, including Ireland [48]. The sugar tax on soft drinks in Ireland is the second highest in Europe, with EUR 0.107 per can of soda [48]. This has led companies to focus stronger on reformulating their products to reduce the amount of sugar whilst supporting the upsurge of sugar-related claims such as “no added sugar”, “sugar free”, or “zero sugar” in order to maintain their soft drink sales [48]. According to an article published in 2020, soft drinks in the European Union have achieved an average 14.6% reduction in added sugars, as well as significant ingredient changes between 2015 and 2019 [49]. A similar increase in the number of claims related to the sugar content on soft drinks was found in a study in Serbia [39].

The comparable trend in “reduced sugar” claims can be seen in the Bread and Bakery product category, where sugar-related claims have multiplied tenfold (from 24% to almost 25%) between 2009 and 2021, representing the desire of consumers to eat less sugar and the actions taken by manufacturers to meet this demand.

Looking at health claims, the prevalence of the usage on packaged food shows the largest changes over time between 2009 and 2021. The number of HC has decreased in each food category, except in the Frozen Fruits and Vegetables and the Fruit Juices and Smoothie categories. The market segment of the latter has experienced steady growth in the years 2017–2021 [50]. This growth may explain the difference in the prevalence of nutrition and health claims between the study conducted in 2009 and the study in 2021. Furthermore, the rise in health claims on prepacked Frozen Fruit and Vegetables is particularly relevant for the Irish market because of the enlarged demand for convenient food over the last 10 years [51]. With increasingly busy lifestyles, other factors, such as cost benefits, prolonged shelf life, and consequently, less food waste, more consumers prefer frozen fruits and vegetables as time-saving alternative for their meals [52].

Overall, the decline in the categories may be related to the Nutrition and Health Claims Regulation and the high degree of difficulty in obtaining approval for a health claim. Indeed, the substantiation process remains challenging for the food industry. Many applications were withdrawn from the process, awaiting more clarity from the authorities before risking negative evaluations due to requirement uncertainties [22]. Notably, the European Food Safety Authority (EFSA) does not clarify what type of studies, nor how many studies, are required to substantiate a claim [53]. In some cases, animal studies or even cell-based studies lead to a positive substantiation; however, other applications will be evaluated unfavourably. Nonetheless, human intervention studies (preferably randomised control trials of the general population) increase the chance of positive evaluations [54]. This generates significant challenges for the food industry because human intervention studies are normally associated with the development of pharmaceutical products. The skills, resources, and know-how to conduct these studies are not typically present in the food industry [55].

Nevertheless, the substantiation of a health claim is considered on a case-by-case basis [15]. Even in the case of a positive opinion from the European Food Safety Authority, the European Commission could still refuse the approval due to social, political, or economic reasons [56]. Overall, it would be fair to say that the prevalence of health claims has decreased, partially due to the long, expensive, and complicated scientific approval process [22].

On the other hand, the harmonised regulation EC 1924/2006 has given consumers a more prominent role in terms of the benchmark of the “average consumer” [33]. The Regulation (NHCR) demands that the consumer is “reasonably well informed and reasonably observant and circumspect” [14]. Consumers must understand the claims and must be protected against any misleading claim [57]. Moreover, the regulation specifies that “the use of nutrition and health claims shall only be permitted if the average consumer can be expected to understand the beneficial effects as expressed in the claim” [14]. However, as mentioned by [33], there is no legal requirement to show that the average consumer understands the claim. The NHCR, therefore, also leads to confusion among consumers, resulting in ambiguous outcomes in the acceptance of nutrition and health claims made on food [32]. Some claims can trigger scepticism and make the consumer believe in false marketing claims, i.e., claims designed for financial benefits rather than actual beneficial health outcomes [58]. In the study published by Pulker CE (2018) [59], 95% of products with health or nutrition claims also included marketing statements highlighting broad health benefits. However, these are not subject to the same level of regulatory scrutiny as health claims, which can mislead consumers who are unable to distinguish between the two. Around 63% of the participants of research conducted in Ireland found nutritional claims confusing [60]. Furthermore, claims are often misinterpreted due to poor nutrition knowledge, thus effecting the trust and credibility of such claims [20,61]. Moreover, the study conducted by Bryla (2020) [61] showed that health and nutrition claims are an important point noted by consumers on food labels; however, the fact of noting this information is also associated with the understanding of information on the label and the perceived credibility. It is important to note that it is still complex to understand how consumers understand claims that depend on a multitude of factors. This is partly the reason why some food business operators choose other methods of differentiating their products whilst complying with the Regulation [62,63]. For example, a company may decide to shift from using health claims to nutrition claims, because the latter have less bureaucratic burden in regard to EC regulation 1924/2006, and hence are easier to use, less costly, and sometimes better understood by the consumer [48,49].

At the same time, nutrition labelling does impact consumers and their food choices [58]. In a literature review performed by Kaur in 2017, 31 publications were analysed, and 20 of these revealed that claims in general (nutrition and health claims) increased purchasing and consumption patterns [64]. The study by Bryla (2020) found that consumers are willing to pay a higher price for food products with health and nutrition claims than for conventional products. For instance, vitamin C and its importance for the human body is well known by the general public. Therefore, the consumer acceptance of these nutrition claims is higher than compared with less well-known compounds [65]. According to Lähteenmäki’s study from 2010, it seems that consumers do not easily accept health claim information unless it is consistent with their knowledge [66].

Additionally, consumers value health claims differently depending on food categories [32]. Health claims on fruit juices or cereal products are more likely to be accepted [32]. This was also reflected in the present study; in fact, most health claims were found on products belonging to the Fruit Juices and Smoothie category, with a percentage of 32.4%.

Similar to the findings by Lalor et al. in 2009, a great number of nutrition claims referred to the product’s fat content, with “low fat” or “reduced fat” being stated on the packs. With the increase in nutrition- and lifestyle-related diseases, the proportion of overweight and obese adults across Europe has almost tripled, resulting Europe’s levels almost matching that of the United States [67]. Due to the perceived link between fat intake and overweight/obesity [68], fat-reduced food products have a wider appeal and food business operators have developed a plethora of products that better fit the consumers’ attempts to have healthier diets and to lose weight. Of particular concern, however, is that these nutrition labels may increase the overconsumption of “low fat” products, leading to higher overall fat intakes in already overweight consumers [69].

Nevertheless, the prevalence of fat claims has stagnated since 2009, whereas the prevalence of low sugar claims has almost doubled (from 11% to 21%). Fisher et al. (2020) [70] identified that in the 20th century, the dietary recommendations were to limit the intake of high-fat food products as much as possible, which resulted in the encouragement of the over-consumption of calories from sweet (high-sugar) foods. However, recent dietary guidelines encourage minimising or eliminating the consumption of refined carbohydrates or sweetened beverages, and the distinction is made between good and bad fats [70]. This may explain further the increase in low/reduced sugar claims in products and the stagnation of fat claims over the last 10 years.

A high portion of nutrition claims refers to the dietary fibre content. Almost 80% of Irish citizens do not eat sufficient amounts of dietary fibre and due to the intake recommendations of 25 g of fibre per day [71], the food industry has increased fibre-related food labelling [72,73]. In the present study, 38% of the nutrition claims were related to fibre, compared with fewer than 2% in 2009. The highest amount of nutrition claims displaying the fibre content were seen in the category Breakfast Cereals followed by Bread and Bakery, which increased from 19% to 28%. Particularly in the category of Bread and Bakery products, fibre-related nutrition claims were the most commonly used, which was also presented in the study conducted in Serbia in 2020 [39].

Although the NHCR aims to promote innovation in the European food sector, some stakeholders view the assessment of health claims as an impeding factor instead [74]. A survey conducted in May 2015 on 50 small- and medium-sized enterprises (SMEs) from 11 European countries reported that 44% of companies had decreased the extent and speed of their innovation [23]. The main reason seems to be the high financial costs associated with the process of applying for claim approval as well as the time that industries need to collate the correct expertise and information to support the claims [23]. In fact, according to estimations, the costs associated with the process of scientific substantiation of health claims range from EUR 4.5 to EUR 7.7 million, which does not include scientific data support or human intervention trials; the entire process may take months or even years [75]. Considering that around 99% of European companies in the field of food and beverages are small- and medium-sized enterprises (SMEs) [23] with limited financial resources, sizeable investment in R&D might not be feasible, thus limiting the applications for health claims [76]. Instead of conducting research on novel product ingredients, food business operators might opt to reformulate their products using well-known ingredients that have already obtained health claim approval [77]. Over time, this approach can act as a disincentive for the food industry to innovate in the area of food science and technology [34].

In general, it can be assumed that, due to various factors including, but not limited to, financial resources, lack of skills, difficulty of compliance with regulations, and the lack of transparency, the NHCR has unfavourably impacted the European food industry’s innovation performance within the research and development sector [78].

Nevertheless, several similar studies investigating the prevalence of health and nutrition claims in supermarkets have been conducted in different countries, both within and outside the European Union.

A survey conducted in five European countries showed that 26% of food products had at least one nutrition claim or health claim, with 64% of those claims being nutritional claims and 29% being health claims [36]. Across the five countries, the United Kingdom had the highest number of nutrition claims (30%). A similar result was observed in the study by Kaur et al. [79], where 32% of the products examined carried out a nutrition claim.

In Serbia, the use of nutrition and health claims is based on the Rulebook adopted in 2018 [64], which fully aligns with the NHCR as Serbia seeks to accede to the EU. It showed that 21.2% of products carried claims (19.4% NC; 8.2% HC) [39]; this number increased compared with 2012, when a similar study had already been conducted [39]. Moreover, this study highlights a significant increase in NC in the Bread and Bakery and Soft Drinks categories, and a decrease in NC for Cheese products, which is similar to the results observed in the current study on the Irish market.

Looking at the prevalence of nutrition and health claims from a more global perspective, a study conducted in Brazil in 2019 found that 41% of packaged food featured claims across different categories [80]. Equivalent to the findings of this study, in Brazil, nutrition claims were prevalent (almost 29%), with 22% of products carrying health claims [80]. Other national differences can be seen; for example, in a study conducted in Australia, almost all products (97% of the examined samples) carried nutrition or health claims, with a higher prevalence of nutrition claims over health claims [81], whereas a study in New Zealand determined that in the food category Breakfast Cereals, almost 96% of products carried nutrition or health claims [82].

It is interesting to note that similar to the study in Brazil, the study conducted in New Zealand determined that 26% of nutrition claims and 7% of health claims were made on “less healthy” food products containing high sugar, saturated fat, or salt contents [83]. Within the food category Breakfast Cereals, almost 65% of the “less healthy” breakfast cereals carried nutrition claims, a much higher percentage than the 17% reporting health claims [82]. When permitting nutrition or health claims on these food products, the overall nutritional value can be obscured, compromising the consumers’ capability to make healthier food choices, while at the same time leading to an increased intake of added sugar and other unfavourable nutrients [80]. Another example, this time in Australia (Pulker CE, 2018), showed that most ultra-processed foods packages included nutrition and health claims, despite the high prevalence of added sugars (95% of products contained added sugars) and being considered less healthy products. In addition, the proportion of inappropriate or inaccurate nutrition claims is considerable, with only 18% of nutrition claims being accurate. 

To counteract these worldwide trends, the European Commission has laid down the adoption of a nutrient profile through Article 4 of the NHCR, stating that food products of certain categories must fulfil strict requirements to be permitted to bear health claims [84]. The documents and guidance on the nutrient profile should have been published at the beginning of 2009. However, due to the complexity and the controversy amongst European Member States, food business operators, and other stakeholders, the implementation has been severely delayed [74]. When it is finally agreed, nutrient profiling may have a significant impact on the use of nutrition and health claims.

Deadlines of the EC Regulation 1924/2006, such as the above-mentioned nutrient profile, have not been met yet; thus, its full impact remains unclear [79]. Therefore, the European Commission conducted research (REFIT) on whether the EC Regulation was still fit for purpose [85]. Although it was concluded that fair competitive conditions amongst the food industry have not been achieved, the regulation is still fit for purpose. Some manufacturers have amended their product recipes to accommodate the possible introduction of nutrient profiles; however, others have yet to do so [26].

Furthermore, the global COVID-19 pandemic that emerged almost two years ago has had a major impact on consumer habits, which may not yet be visible in supermarkets but may be noticeable in the upcoming years.

This study is subject to limitations. Due to limited resources and time constraints, only three supermarkets in Ireland were considered for the data collection. Although a variety of supermarkets in size and popularity among the Irish population was taken into account, as well as the fact that the supermarket chains were available nationwide, the data collected do not entirely represent all the food available on the Irish market. The day of data collection will always have an impact on the results, because some products may have been absent from the shelves or out of stock at the time of collection. Furthermore, the comparison with the results of the study by Lalor et al. (2009) [16] is only partially possible, because the initial study included more categories than the eight chosen for the present investigation. It would be interesting to conduct the survey on the last remaining food categories to enable a complete overall view on all food categories.

## 5. Conclusions

This study was conducted to assess the trend over time in the prevalence of nutrition and health claims on packaged food products available on the Irish market. With the implementation of the EC Regulation 1924/2006 15 years ago, it is important to observe the impact of time on the use of both nutrition and health claims on the Irish market in order to provide a comprehensive picture of the outcomes and usage of the harmonised legislation across Member States.

The present study has shown the changes in the use of NHC between 2009 and 2021 on eight food product categories. Overall, the proportion of nutrition claims has remained broadly stable; however, it is possible to highlight differences between the categories. The use of claims and the type of nutrients involved in the claims may reflect the current market trend, which varies from category to category.

In addition, the proportion of health claims on food products has decreased. Indeed, across Member States, financial obstacles and legislative ambiguities continue to prevent companies from committing to the expensive and time-intensive health claim application process. Key challenges of EC Regulation 1924/2006 are arguably the scientific substantiation process and the influence of its complexity on food innovation. The financial burden and lack of expertise in the industry are key restrictive factors.

Although the REFIT evaluation conducted in 2020 found that the EC Regulation 1924/2006 is still fit for purpose, it appears evident that not all objectives have been achieved, namely, the establishment of the nutrient profile or the requirements for the assessment criteria for health claims on plant-based products. Food business operators must deliberate carefully whether, current circumstances notwithstanding, the use of nutrition and health claims is still beneficial for them. These decisions, on a systematic level, affect and will continue affecting the prevalence of nutrition and health claims on the European market.

Ultimately, the results from this study will help to provide decision- and policymakers, as well as the food industry, with relevant information on the prevalence of nutrition and health claims over time. The data collected could contribute to supporting the development of policy amendments and/or marketing strategies that empower consumers to make better and healthier choices, whilst also ensuring the adequate communication of beneficial health-related effects of food products or ingredients.

## Figures and Tables

**Figure 1 foods-11-02789-f001:**
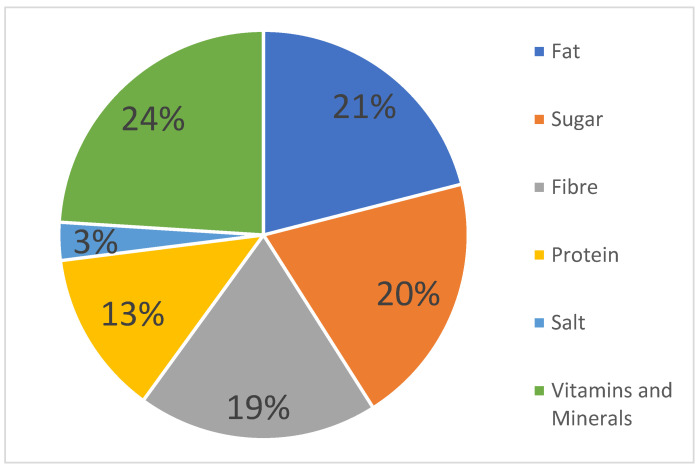
Proportion of nutrients referred to in nutrition claims.

**Table 1 foods-11-02789-t001:** Explanation and examples of nutrition and health claims as provided for in Regulation (EC) 1924/2006 [20] Reproduced from [20] with permission from Taylor & Francis Online, 2022 (www.tandfonline.com, accessed on 5 April 2022).

			Regulation (EC) No 1924/2006			
Type of Claim	Nutrition Claim		Health Claims			
Name	Content Claim	Comparative Claim	Function Claims		Risk Reduction Claim	Health Claims referring to child development
Parameter			Based on generally accepted scientific evidence	Based on newly developed scientific data	Based on newly developed scientific data	Based on newly developed scientific data
Reference	Art. 8	Art. 9	Art. 13.1	Art. 13.5	Art. 14 (a)	Art. 14 (b)
Example	“Source of Vitamin C”	“Reduced sugar”	“Vitamin C increases iron absorption”	“Cocoa flavanols help maintain endothelium dependent vasodilation which contributes to normal blood flow”	“Plant sterols have been shown to lower/reduce blood cholesterol. High cholesterol is a risk factor in the development of coronary heart disease”	“Calcium is needed for a normal growth and development of bones in children”

**Table 2 foods-11-02789-t002:** Number of assessed health claims and number of authorised health claims in the EU (May 2022) [27].

Type of Claim	Number of Assessed Claims	Number of Authorised Claims
Reduction of disease risk claims	41	14
Claims referring to children’s development and health	57	12
Health claims based on newly developed scientific evidence	136	5
Health claims supported by generally accepted scientific evidence	4637	229

**Table 3 foods-11-02789-t003:** Number of products per category.

Food Categories	Number of Products
Breakfast cereals	223
Yoghurt and yoghurt drinks	296
Fruit Juices and smoothies	170
Frozen fruits and vegetables	81
Breads and bakery products	293
Pastas and rice	110
Cheeses	286
Soft drinks	177
Total	1636

**Table 4 foods-11-02789-t004:** Proportions of products with nutrition claims, health claims, or both.

	*n*	%
With nutrition and health claim	150	9.2
Nutrition claims	863	52.8
Health Claims	172	10.5
With nutrition or health claim	885	54.1
Total	1636	100

**Table 5 foods-11-02789-t005:** Prevalence of nutrition and health claims in 2021 by food category.

	Products with ≥ 1 Nutrition Claim		Products with ≥ Health Claim		Total Number of Products Examined
	*n*	%	*n*	%	
Breakfast Cereals	181	81.2	46	20.6	223
Yoghurt and Yoghurt drinks	184	62.2	53	17.9	296
Fruit Juices and Smoothies	102	60	54	31.8	170
Frozen Fruit and Vegetables	46	56.8	9	11.1	81
Breads and Bakery Products	135	46.1	1	0.3	293
Pasta and Rice	30	27.3	0	0	110
Cheeses	82	28.7	9	3.1	286
Soft Drinks	103	58.2	0	0	177
Total	863	52.8	172	10.5	1636

**Table 6 foods-11-02789-t006:** Number and types of nutrients referred to in nutrition claims.

Food Categories	Nutrient						
	Fat	Sugar	Fibre	Protein	Salt	Vitamins and Minerals	Total
Breakfast cereals	43	40	147	16	38	59	343
Yoghurt and yoghurt drinks	121	55	9	95	0	81	361
Fruit Juices and smoothies	0	53	11	1	0	76	141
Frozen Fruit and vegetables	8	2	11	5	0	28	54
Bread and bakery products	80	78	89	34	1	22	304
Pasta and Rice	9	0	18	16	4	0	47
Cheeses	49	6	0	33	0	47	135
Soft Drinks	4	61	0	0	0	49	114
Total	314	295	285	200	43	362	1499

**Table 7 foods-11-02789-t007:** Evolution of the proportion of claims by food category between 2009 and 2021.

Food Category	NC Observed in 2009 (Lalor et al.)		NC Observed in Current Study 2021		NC *p*-Value Chi-Squared Test	HC Observed in 2009 (Lalor et al.)		HC Observed in Current Study 2021		HC *p*-Value Chi-Squared Test
	*n* with NC	%	*n* with NC	%	*p*	*n* with HC	%	*n* with HC	%	*p*
Breakfast Cereals	140	85	181	81.2	0.415	70	42	46	20.6	0.001
Yoghurt and Yoghurt Drinks	73	55	184	62.2	0.181	66	50	53	17.9	0.001
Fruit Juices and Smoothies	34	83	102	60	0.006	12	29	54	17.9	0.001
Frozen Fruit and Vegetable	42	86	46	56.8	0.001	0	0	9	11.1	0.016
Breads and Bakery	75	46	135	46.1	0.944	13	8	1	0.3	0.001
Pasta and Rice	27	31	30	27.3	0.563	27	31	0	0	0.001
Cheese	103	41	82	28.7	0.002	39	16	9	3.1	0.001
Soft Drinks	102	45	103	58.2	0.007	14	6	0	0	0.001
Total products all categories	596	31.7	863	52.8	0.736	241	17.8	172	10.5	0.001

## Data Availability

Not applicable.

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
