# Peer review of "The Impact of Time on Nutrition and Health Claims on the Irish Marketplace"

_foods, 2022, doi:10.3390/foods11182789_

Round 1
Reviewer 1 Report
After reading the manuscript "The impact of time on nutrition and health claims on the Irish 2 marketplace", I realized that the manuscript showed in some parts the scientific rigour wanted, but in other parts I have missed it. The authors have presented critical evaluation only in some paragraphs. The references are not exactly current, besides the objective could be presented earlier.
Thats why I have written some suggestions below in an attempt to improve the paper.
L. 24- define BSE, please, some readers may not know.
L.30- Figures ?
L.60- Please, insert more clearly what the rationale and objectives of the paper are.
L.64- 93- So sorry, but the articles part got a little tiresome. Please, think of a strategy to get the reader more involved, maybe in the table, figure...
L.66- 67- I keep wondering how to separate Nutrition from Health. Hard task...
L.98- Please, improve Table 1
L.118- In figure 1, Nutrition claims has become very loose, without much information. In Health Claims you should also comment more about what happens after rejection. Can a new attempt be requested after some time?
L.158- wow! Only after 5 pages we find the rationale and the objectives. I suggest improving this part, it is unusual.
L.168- with these 3 evaluated markets, did you succeed in contemplating different social classes and regions of the country? upper and lower classes? rural and urban areas?
L.218- It got very repetitive (3x) - 1636
L.222- How were the foods selected that culminated in table 3? For example, why were breads and bakery (293) products evaluated? How was this stratification performed. Insert this information in Material and methods ? Random selection ?
L. 224- Why is table 4 in italics ? Placing the total (n) at the bottom is the most appropriate.
L.244- Table 6 deserved a wider discussion about these findings. For example, not having Vitamins and minerals (n=0) in pasta & rice is so strange. Salt as well, deserved a deep discussion, and so on.
L.246- Also, "Nutrients" doesn't sound appropriate to me, as fibres are not considered nutrients because they are not absorbed. Review . Another question, separate "salt" from "Minerals", can be a bias in this outcome ?
L.251- Figure 2 in colour would be more interesting.
L.285- Improve table 7, please. It seems to me that "observed in" and "by" could be omitted. Put "n" and "%" in bold to make it more highlighted.
Reviewer 2 Report
This is a good quality manuscript. The research problem is relevant and important. The applied methodology is correct. The results are presented clearly. They are interpreted and discussed in the right way.
Please calculate Chi2 for each product category separately comparing the prevalence of: 1) nutrition claims in 2009 and 2021, b) health claims in 2009 and 2021.
Please refer to:
Bryła P., Who reads food labels? Selected predictors of consumer interest in front-of-package and back-of-package labels during and after the purchase, Nutrients, 2020, Vol. 12, 2605. https://doi.org/10.3390/nu12092605.
Pulker, C. E., Scott, J. A., & Pollard, C. M. (2018). Ultra-processed family foods in Australia: nutrition claims, health claims and marketing techniques. Public Health Nutrition, 21(1), 38-48.
line 19 - these
29 - deaths
45 - provided
54 - State
93 - [13.16.17]
Table 1 - Claims
In Table 4, the last row is misleading. Shouldn't it be 52.8+10.5=63.3%?
In Table 5 - points instead of commas
278 - difference
366 - applications have been
393 - affecting
426 - possible,
438 - Bakery,
472 - is based
Reviewer 3 Report
The topic is interesting and has wider practical and theoretical implications. The authors have put their best efforts to execute this paper. However, I have the following reservations and suggestions for the sake of improvement of the undertaken study:
Abstract
The logical sequence of the abstract should be as 1) objectives, 2) methodology, 3) Findings, 4) conclusion and 5) implications. Thus, the authors should also rewrite the abstract in this sequence. The authors have mentioned the important statistical techniques, which they have used in their research. Finally, important implications should be mentioned at the end of the abstract.
Introduction
The authors did not establish the motivation, significance, and novelty of the undertaken study. The authors suggested improving this important factor in the "Introduction" section. The authors should have described the proper background and significance of this study with respect to the Irish market. Why did the authors take only the Irish market? The scope of the study has narrowed, please justify. Additionally, the authors did not establish the significance, novelty, and even objectives of the paper. Please take care of these factors, these are the essence of the scientific paper.
The authors should use a separate heading for the Review of literature, and establish the importance and significance by identifying the gap from the previous literature. The literature should always be presented in an audit form i.e. critically analyzed & identify the gap, and linked with the objectives of the current paper.
Material and methods
The material and methods section should need further elaboration because the authors have described the entire methodology in one single heading. The authors should write the first subheading “Research Design”, the second, sub-heading should be the data collection method, lastly sub-heading of sampling strategy and why selected this sample size? and finally, the authors should write in detail the statistical estimation techniques and how it satisfies the objectives of the research.The demography of respondents needs a separate sub-heading.
Results
Since the authors used several statistical techniques for their analyses, there is a lack of explanation for these results. The authors are requested to explain results in an interpretation manner not just in statistical numbers. The authors should present the missing parts and sequence the sub-headings as used in the statistical methods.
Discussions & conclusions
The discussions section provides the opportunity for the authors to sell their idea to the readers. The discussions section should be complemented with the previous literature. The authors should compare their results with the previous literature on whether they are consistent and why? If they are not consistent with the previous literature then what is the reason and how it should be contributed to the body of knowledge. The conclusion needs further elaboration. The conclusion is always one step ahead of the findings. The conclusion should not be the repetition of results and findings only.
Theoretical, practical, and societal implications
The practical, theoretical, and societal implications should be discussed in separated sub-headings after the conclusion, and in the light of the conclusion and discussions. However, the authors have discussed the practical and theoretical implications in the last paragraph of the paper.
In the end, I will emphasize the authors should add the limitations and potential areas for future studies at the end of the paper. The minor spelling and grammatical mistakes should be corrected throughout the paper.
Round 2
Reviewer 1 Report
Dear authors,
After another evaluation of the manuscript, I see a great improvement in the quality of the paper. The authors have accepted almost all of my requests.
In figure 1 I didn't understand you mentioned Nuala Collins, (FSAI) in the response letter, but the reference is 24.
I don't understand. This figure is adapted from which author?
"These eight categories were chosen to match those in the study published 12 years earlier by Lalor et al. (2009), to enable an accurate comparison. L. 188 – 189." - Insert this explanation you provided me in item 3.1.1
Page11- Lalor without the year. Check, please.
Author Response
- Regarding the reference for Fig. 1, the manuscript is correct. The reference to Nuala Collins that appeared in the reply to the reviewer, was made in error - please disregard.
- This sentence: "These eight categories were chosen to match those in the study published 12 years earlier by Lalor et al (2009) to enable an accurate comparison." has been added to the manuscript as recommended
- Lalor reference on page 11 has been checked.
thanks!
Reviewer 3 Report
The authors have addressed all the comments and made changes in the Introduction, Review of Literature, Material & Methods, Results, discussions, and conclusions.
Author Response
No recommendations were received on this 2nd review.
Thanks!